# Characterisation of a Hepatitis C Virus Subtype 2a Cluster in Scottish PWID with a Suboptimal Response to Glecaprevir/Pibrentasvir Treatment

**DOI:** 10.3390/v14081678

**Published:** 2022-07-29

**Authors:** Rajiv Shah, Stephen T. Barclay, Erica S. Peters, Ray Fox, Rory Gunson, Amanda Bradley-Stewart, Samantha J. Shepherd, Alasdair MacLean, Lily Tong, Vera Jannie Elisabeth van Vliet, Michael Ngan Chiu Bong, Ana Filipe, Emma C. Thomson, Chris Davis

**Affiliations:** 1Thomson Group, College of Medical, Veterinary & Life Sciences, MRC-University of Glasgow Centre for Virus Research, Glasgow G61 1QH, UK; rory.gunson@ggc.scot.nhs.uk (R.G.); lily.tong@glasgow.ac.uk (L.T.); vera.vliet@hotmail.com (V.J.E.v.V.); nganchiubong@gmail.com (M.N.C.B.); ana.dasilvafilipe@glasgow.ac.uk (A.F.); emma.thomson@glasgow.ac.uk (E.C.T.); 2NHS Greater Glasgow & Clyde, Departments of Hepatology and Virology, Glasgow Royal Infirmary, Glasgow G4 0SF, UK; stephen.barclay@ggc.scot.nhs.uk (S.T.B.); peterer956@ggc.scot.nhs.uk (E.S.P.); ray.fox2@ggc.scot.nhs.uk (R.F.); amanda.bradley-stewart@nhs.scot (A.B.-S.); samantha.shepherd@ggc.scot.nhs.uk (S.J.S.); alasdair.maclean@ggc.scot.nhs.uk (A.M.)

**Keywords:** hepatitis C virus, direct-acting antivirals, glecaprevir, pibrentasvir, sustained virological response, resistance associated substitutions, phylogenenetics, subgenomic replicons

## Abstract

Direct-acting antivirals (DAAs) have revolutionised the treatment of Hepatitis C virus (HCV), allowing the World Health Organisation (WHO) to set a target of eliminating HCV by 2030. In this study we aimed to investigate glecaprevir and pibrentasvir (GP) treatment outcomes in a cohort of patients with genotype 2a infection. Methods: Clinical data and plasma samples were collected in NHS Greater Glasgow & Clyde. Next generation whole genome sequencing and replicon assays were carried out at the MRC-University of Glasgow Centre for Virus Research. Results: 132 cases infected with genotype 2a HCV were identified. The SVR rate for this group was 91% (112/123) following treatment with GP. An NS5A polymorphism, L31M, was detected in all cases of g2a infection, and L31M+R353K in individuals that failed treatment. The results showed that R353K was present in 90% of individuals in the Glasgow genotype 2a phylogenetic cluster but in less than 5% of all HCV subtype 2a published sequences. In vitro efficacy of pibrentasvir against sub-genomic replicon constructs containing these mutations showed a 2-fold increase in IC_50_ compared to wildtype. Conclusion: This study describes a cluster of HCV genotype 2a infection associated with a lower-than-expected SVR rate following GP treatment in association with the NS5A mutations L31M+R353K.

## 1. Introduction

An estimated 58 million people in the world have chronic Hepatitis C virus (HCV) infection, of whom one third will develop liver cirrhosis with associated risks of decompensated liver disease and hepatocellular carcinoma (HCC). In 2015, the WHO estimated that there were 1.75 million new infections, predominantly due to unsafe health-care procedures and injecting drug use [1]. The development of interferon-free, all oral regimens of direct-acting antivirals (DAAs) has revolutionised HCV treatment. Indeed, the global elimination of HCV has generated much interest and appears to be a realistic goal, as evidenced by the World Health Organisation (WHO) 2030 target [2].

While the rates of sustained virological response 12 weeks after treatment (SVR12) are as high as 95% [3], treatment failure occurs in a small number of individuals. Genetic surveillance of cases of treatment failure is useful, to characterise the emergence of viral drug resistance [4], a risk that occurs due to the extensive natural genetic diversity of HCV. Under drug pressure, mutations within the targeted viral proteins NS3, NS5A and NS5B may emerge, giving rise to treatment failure [5]. The NS3-inhibitors, including glecaprevir, bind to the NS3 protease catalytic site, preventing posttranslational processing of the viral polyprotein and the release of proteins that are essential to produce viral particles. The NS5A inhibitors such as pibrentasvir and NS5B RNA polymerase inhibitors impair viral replication and the production of viral particles. Treatment guidelines recommend combination DAA therapy [6,7,8] to reduce the risk of such resistance developing. However, 5–10% of individuals may fail to achieve an SVR12, with the highest risk in individuals who are cirrhotic, require retreatment, are non-adherent or are infected with diverse HCV subtypes such as 1l, 4r and 6b, usually acquired in Asia or Sub-Saharan Africa [9].

Here, we describe a cluster of individuals infected with genotype 2a with a suboptimal response to treatment with glecaprevir/pibrentasvir (G/P), a regimen previously reported to have a high genetic barrier to resistance. We also investigate a subset of individuals from this group who failed treatment. Resistance was associated with the combination of L31M and R353K mutations within NS5A.

## 2. Methods

### 2.1. Clinical Cohort and Samples

Demographic and clinical data on individuals with HCV genotype 2 infection, managed within NHS Greater Glasgow and Clyde (GGC), and treated with G/P, were collected between 2013 and 2019. Plasma samples from fifteen of these individuals, including two that failed treatment, were available for whole HCV genome sequencing. Liver cirrhosis was defined by a transient elastography score of greater than 12 kPa. Fisher’s exact tests were conducted for categorical variables and Wilcoxon Rank Sum test for continuous variables. Ethical approval was granted by the National Research Ethics Service Committee, East Midlands (11/EM/0323) and from the West of Scotland Research Ethics Committee (12/WS/0002).

### 2.2. HCV RNA Quantification

HCV viral load testing was carried out at the West of Scotland Specialist Virology Centre, Glasgow Royal Infirmary. Viral loads were determined by the Abbott Alinity m in vitro realtime reverse transcriptase test (Abbott Molecular, IL, USA). It has a limit of detection (LOD) of 12 International Units per millilitre (IU/mL) and a quantitation range of 12 to 2 × 10^6^ IU/mL. Validation shows all HCV genotypes are detected and quantitated equally.

### 2.3. Library Preparation and Next Generation Sequencing

Extraction of viral RNA from the plasma samples (200 µL) was performed with the Agencourt RNAdvance blood kit (Beckman Coulter, Brea, CA, USA). The RNA was then eluted in nuclease free water (11 µL) and reverse transcribed with Superscript III (Invitrogen, MA, USA), using random hexamers. NEB Second Strand Synthesis kit (New England Biolabs, MA, USA) and KAPA Library Prep kit (KAPA Biosystems, MA, USA) were used for library preparation. Samples were indexed using NEBNext Multiplex Oligos for Illumina (New England Biolabs, MA, USA). Qubit (Thermo Fisher, MA, USA) and TapeStation (Agilent, CA, USA) were then used to quantify and check the quality of the amplified DNA. The libraries were then pooled at similar molar concentrations and target enrichment (TE) was performed using NimbleGen SeqCap EZ system RNA probes (Roche, CH) as previously described [10]. Samples were then sequenced on the Illumina MiSeq platform (Illumina, CA, USA) using a v3 MiSeq Reagent Kit (Illumina, Illumina Centre, Cambridge, UK).

### 2.4. Bioinformatic Analyses

NGS data were analysed using an in-house Unix-based analysis and genotyping pipeline. In brief, raw sequence reads in the form of fastq files were cleaned and quality checked and genotyped using a KMER based approach. A BLAST guided programme, Tanoti, was then used to map the sequences to the relevant reference sequences in order to generate consensus genomes. Whole genomes were accepted, where 90% of the genome was covered, and then aligned using MAFFT (multiple alignment program for amino acid or nucleotide sequences) Version 7.313 [11]. Finally, a maximum likelihood phylogenetic tree was constructed using RAxML [12]. The tree was constructed using a general time reversible nucleotide substitution model with 1000 bootstraps. Genetic distance between the sequences was calculated using MEGA X [13], and using a bootstrap method for variance estimation. HCV GLUE software (http://hcv.glue.cvr.ac.uk, accessed 13 June 2021) was used to evaluate and analyse resistance-associated mutations present in the sequenced samples [14]. Aligned, consensus amino acid sequences were inspected within NS3 and NS5A genes to identify mutations unique to HCV from the individuals who failed treatment. ClusterPicker [15] was used to define closely related sequence clusters that were less than 10% different to each other across the whole genome. Whole genomes generated as part of this study have been submitted to GenBank (www.ncbi.nlm.nih.gov/genbank/, accessed on 18 July 2022). Appendix B lists the assigned accession numbers.

### 2.5. Sub-Genomic Replicon (SGR) Constructs

A sub-genomic replicon (pJFH-1) encoding a Gaussia luciferase gene [16] was modified to encode the NS5A amino acid substitutions identified as unique to the virus infecting individuals who failed treatment. The NS5A constructs containing polymorphisms that have previously been associated with resistance to pibrentasvir were also synthesised and used as controls. These polymorphisms included F28F+L31I and P29S+K30G.The background plasmid sequence, spanning non-structural proteins NS3 to NS5B of HCV, was that of JFH-1, which is a subtype 2a HCV sequence that was isolated from a Japanese individual with fulminant hepatitis [17]. This strain of HCV is known to replicate well in cell culture and is sensitive to DAAs. The constructs were manufactured by Invitrogen (ThermoFisher Scientific, UK) using a site-directed mutagenesis system. A pJFH-1/GND replicon, containing a self-inactivating mutation in the NS5B gene, was used as the negative control.

### 2.6. RNA Transcription and Electroporation

Replicon constructs were linearised with Xba1 (New England Biolabs, MA, USA) and then purified (Monarch DNA gel extraction kit, New England Biolabs, MA, USA). Purified DNA was then used as a template for RNA transcription (T7 RiboMAX Express Large Scale RNA Production System, Promega, WI, USA) followed by RNA purification (RNeasy Mini Kit, Qiagen, Hilden, Germany). RNA was electroporated (270V, 950 capacitance, ∞ resistance) in 4mm cuvettes (Molecular BioProducts, ThermoFisher Scientific, UK) at 2 × 10^5^ cells per reaction. Electroporated cells were immediately chilled on ice to rest for a few minutes before being resuspended in 5mls of 10% foetal bovine serum DMEM (Gibco, ThermoFisher Scientific, UK). Cells were seeded into 96-well plates and incubated at 37 °C for 4 h.

### 2.7. Replication Capacity of SGR Constructs

To ensure the SGR constructs were efficiently replicating we collected supernatant at 4, 24, 48 and 72 h for luminescence reading. We then compared the replication efficiency to that of wildtype (JFH-1) by calculating replication capacity. This was calculated using the luminescence readings in the following formula: (Mutant 72 h/Mutant 4 h)/(JFH-1 72 h/JFH-1 4 h).

### 2.8. Sub-Genomic Replication Inhibition Assay

The 96-well plates were seeded with electroporated cells as mentioned above and then treated with Pibrentasvir (Cayman Chemical, MI, USA, CAS No. 1353900-92-1) in serial dilutions. Cells were incubated with drug for 72 h before culture medium was removed and luciferase assay performed (Pierce™ Gaussia Luciferase Flash Assay Kit, ThermoFisher Scientific, UK). Relative light units (RLU) were calculated as the 72 h read divided by the 4 h read for each well. Maximum responses were set as the mean RLU of wells untreated with drug and the remaining RLU were normalised to this value and expressed as percentages. Drug response curves and IC50 values were calculated using non-linear regression (GraphPad Prism 9 software).

## 3. Results

### 3.1. Individual Characteristics

Between 2013 and 2019, 132 individuals infected with HCV genotype 2a were treated with G/P therapy. The median age was 48 years and 72% (95/132) were male. Outcome data were missing for nine individuals. Of those remaining, 91% (112/123) achieved SVR12. Table 1 summarises the demographic and clinical characteristics of this group. None of the individuals who failed treatment had liver cirrhosis. One individual developed HCC while being successfully treated with DAAs. None of the individuals who failed treatment had HCC prior to or post DAA therapy. Route of transmission was known for 76% (94/123), and of these, 70% (86/123) were people who inject drugs (PWID).

### 3.2. Phylogenetic Analysis

A maximum likelihood tree was constructed with reference strains from different genotype 2 subtypes. Appendix A shows the full genotype 2 clade and reveals that the circulating HCV subtypes in the clinical cohort included both subtypes 2a and 2b. The subtype 2a cluster that includes individuals P19 and P20, who failed treatment, is shown in Figure 1. This cluster of sequences was distinct to all other sequences withing the subtype 2a clade, as analysed in ClusterPicker (https://hiv.bio.ed.ac.uk/software.html, accessed 19 May 2022). Individuals P6 and P10 were successfully treated with (G/P); P4, P7, P8 and P9 were not treated with (G/P). We noted that there was a sequence from an individual in Edinburgh from 2013 and two other sequences from UK individuals (sequence accession numbers KY320321 and KY620322), also from 2013, that formed part of this genotype 2a cluster, suggesting circulation of this subtype 2a clade since at least 2013 outside of the NHS GG&C area.

### 3.3. Resistance-Associated Substitutions

Resistance-associated substitutions (RASs) were analysed using HCV GLUE [14]. None of the known NS3 RASs (A156T/V, D168E/V) previously associated with glecaprevir resistance in HCV genotype 2a [19] were noted in the individuals who failed therapy. The known NS5A RAS L/M31M was identified in HCV sequences from P19 and P20 as well as the other sequences in the same subtype 2a cluster. The proportion of sequences with leucine and methionine at position 31 was 15% (22/147) and 85% (125/147), respectively, in all genotype 2a sequences from GenBank, curated by HCV GLUE. We compared available whole genome sequences from the individuals that failed (G/P) treatment (P19 and P20) to HCV sequences from individuals that were successfully treated (P6 and P10) and noted that NS5A polymorphisms R353K and P407L were present in P19 and P20, but not in P10. These were taken forward for in vitro resistance analysis. We next looked to see if the two NS5A polymorphisms were present in the other sequences within the subtype 2a cluster. Out of the remaining 10 sequences, 9 (90%) and 7 (70%) had the R353K and P407L polymorphisms, respectively. Finally, we checked to see how common these two NS5A polymorphisms were in all subtype 2a sequences by analysing an alignment of all subtype 2a NS5A sequences from HCV GLUE. The proportion of sequences that had the R353K and P407L polymorphisms were 11/257 (4.3%) and 138/257 (53.7%), respectively.

### 3.4. Replication Capacities of Sub-Genomic Replicon (SGR) Constructs

We tested the efficacy of pibrentasvir against SGRs harbouring the above-mentioned polymorphisms in NS5A alongside L31M polymorphism which was observed in P19 and P20 as well as P6 and P10. We first checked the replication capacities of the SGR constructs, the results of which are shown in Figure 2. The control SGR constructs (harbouring polymorphisms F28S+L31I and P29S+K30G) had greater replication capacities (median replication capacities of 1.3 and 1.2, respectively) than JFH-1 (wildtype). The same effect was seen with the SGR construct harbouring mutation P407L in NS5A (replication capacity of 1.4). The SGR construct with mutation R353K had a similar replication capacity to JFH-1.

### 3.5. In Vitro Efficacy of Pibrentasvir

The efficacy of pibrentasvir in replicons expressing R353K was tested with a serial dilution of pibrentasvir, in order to determine mean fold change in IC_50_. Figure 3 shows the effect of polymorphisms R353K and P407L when compared to JFH-1. The IC_50_ value for R353K (IC50 = 2.42 × 10^−3^, 95% CI = 9.54 × 10^−4^ to 6.546 × 10^−3^, R^2^ = 90.5%) was 2-fold higher than the IC_50_ value for JFH-1 (IC50 = 9.24 × 10^−4^, 95% CI = 6.68 × 10^−4^ to 1.28 × 10^−3^, R^2^ = 99%). The IC_50_ value for P407L was comparable to that of JFH-1.

## 4. Discussion

In this study, we describe a cluster of HCV subtype 2a infection in 123 individuals who were treated with G/P and achieved a lower than optimal SVR rate of 91%. The EASL guidelines do not routinely recommend baseline HCV resistance testing except in cases of acquisition of infection in areas harbouring subtypes that have a low barrier to resistance [20]. This study aimed to identify mutations associated with lack of SVR12 following G/P therapy, which is considered a pan-genotypic regimen, after an increase in cases of relapse following this treatment in a single health board. Both drugs are also second generation in their respective drug classes and have a higher barrier to resistance than the first-generation drugs. Phylogenetic analysis of the cohort revealed a cluster with more than 90% similarity to each other across the whole genome, suggesting a localised transmission of subtype 2a infection in individuals in Glasgow and Edinburgh. This clade has likely been circulating in Scotland for at least 10 years.

The SVR rate of 91% in this group of individuals is slightly lower than reported in clinical trials of G/P combination treatment [21,22,23]. We identified a combination of two mutations within NS5A (R353K and P407L) in individuals that failed therapy; R353K in combination with L31M is associated with low–moderate resistance to G/P therapy while P407L is associated with an increase in replication capacity. The R353K polymorphism was present in 79% (11/14) of the sequences in this cluster, but only in 4% (11/257) of all published subtype 2a sequences that include the NS5A region. When R353K is combined with L31M it has a 2-fold higher IC_50_ value than wildtype when tested in an in vitro SGR replicon assay. The most common previously reported polymorphisms associated with resistance to DAAs in genotype 2 are at sites 24, 28–31, 92 and 93 [24,25,26,27,28]. These are all sites within domain 1 of NS5A, which is primarily involved in RNA replication. Crystal structures have been solved for NS5A domain 1 [29]. Site 353 is an unusual site to be associated with resistance as it occurs within the linker region between domains 2 and 3, closer to domain 3, which is involved in virion assembly [30,31] and shows in vitro RNA binding capacity [32]. It may have been undetected in more limited sequencing studies that did not employ whole genome sequencing methods.

While this study was not designed, nor powered, to look at clinical or demographic factors associated with treatment failure, we noted that males were over-represented in the treatment failure group. All 11 individuals that failed treatment were male and 71% (80/112) of individuals that achieved SVR were male (Fisher’s exact test *p* = 0.0648).

In this study, we report a lower-than-expected SVR rate among a group of individuals with chronic genotype 2a HCV infection treated with G/P. A polymorphism at site 353 in NS5A was found to be more common in this strain than in other subtype 2a sequences and showed an increase in IC_50_ in an in vitro SGR replicon assay when challenged with pibrentasvir. The combination of L31M and R353K was present in both cases of treatment failure. The P407L mutation also detected in this cluster is associated with an increase in replication fitness.

## Figures and Tables

**Figure 1 viruses-14-01678-f001:**
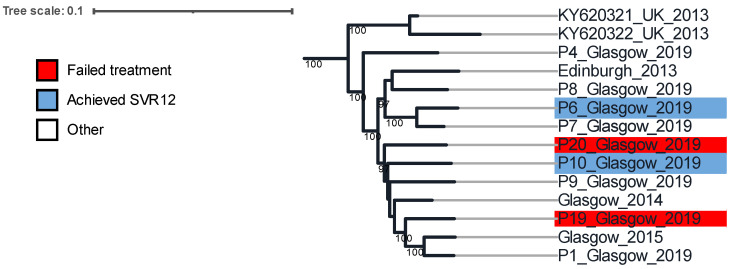
Maximum likelihood tree of whole nucleotide HCV genomes constructed with 1000 bootstrap replicates. This tree shows the Glasgow subtype 2a phylogenetic cluster that includes treatment failures P19 and P20, and P6 and P10 who were both successfully treated. Non-highlighted tree branches represent other sequences, from Scotland, in this 2a cluster, and two HCV sequences downloaded from GenBank [18], also from UK.

**Figure 2 viruses-14-01678-f002:**
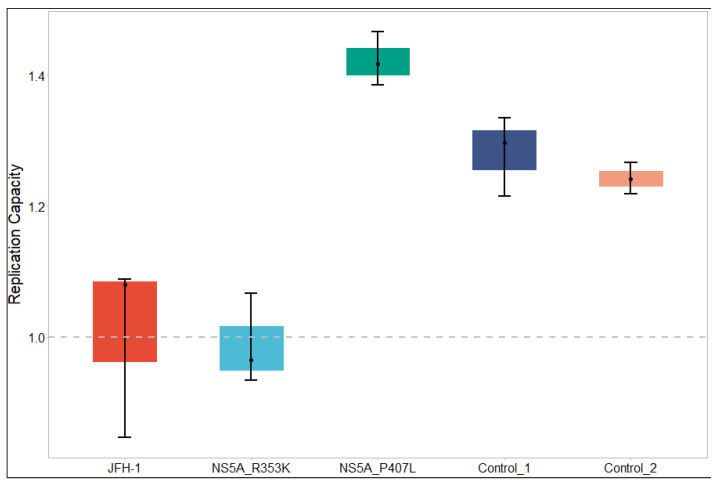
Replication capacities of JFH-1 (wildtype), SGR constructs harbouring polymorphisms unique to P19 and P20 (R353K and P407L), and controls. Control_1 harbours the polymorphisms F28S and L31I. Control_2 harbours the polymorphisms P29S and K30G.

**Figure 3 viruses-14-01678-f003:**
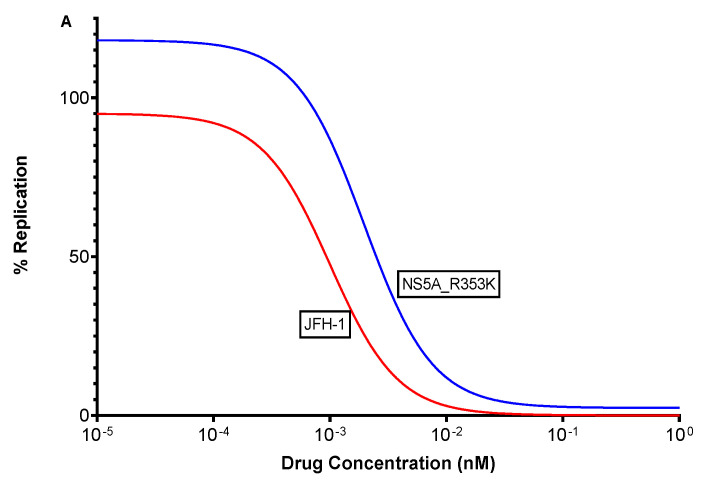
IC50 drug response curves shown for SGR constructs. The constructs harbouring NS5A_R353K is shown in panel (**A**) and NS5A_P407L in panel (**B**).

**Table 1 viruses-14-01678-t001:** Summary of demographic and clinical characteristics of HCV genotype 2 infected individuals treated with G/P between 2013 and 2019. * SVR = sustained virological response, ** IDU = injecting drug use. *** This includes other transmission risks including sexual transmission, tattoos and transmission through blood product administration. ^†^ Fisher’s exact test used. ^††^ Wilcoxon rank sum test used.

	Treatment Outcome (*N* = 123)		*p* Value
	SVR * (*N* = 112)	Failure (*N* = 11)	
**Gender**			
Male	80 (71%)	11 (100%)	
Female	32(29%)	0 (0%)	0.0648 ^†^
**Mean Age (standard deviation)**	48.8 (±10)	47.5 (±10)	0.8731 ^††^
**Mean HCV Viral Load (IU/mL)**	5.88 (±0.9)	5.98 (±0.7)	0.8593 ^††^
**HIV Status**			
HIV negative	107 (96%)	11 (100%)	1 ^†^
HIV positive	5 (4%)	0 (0%)	
**Liver Status**			
Non-Cirrhotic	96 (86%)	11 (100%)	0.356 ^†^
Cirrhosis (>12KPa on transient elastography)	16 (14%)	0 (0%)	
**Transmission**			
IDU **	78 (69%)	8 (73%)	0.6724 ^†^
Other ***	10 (9%)	0 (0%)	
Unknown	24 (22%)	3 (27%)	

## Data Availability

Genomes have been uploaded to GenBank. Accession numbers are listed in Appendix B, Table A1. Aggregated clinical data is presented in the paper but individual data are not available due to ethical restrictions on the use of identifiable information.

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
