# Peer review of "Characterisation of a Hepatitis C Virus Subtype 2a Cluster in Scottish PWID with a Suboptimal Response to Glecaprevir/Pibrentasvir Treatment"

_viruses, 2022, doi:10.3390/v14081678_

Round 1
Reviewer 1 Report
It is an important study which characterizes HCV subtype 2a associated with treatment (glecaprevir/pibrentasvir) failure in Scotland. The authors reported that 91% achieved SVR, and they identified the presence of NS5A L31M and R353K mutations in case of treatment failure. In vitro assays showed that these mutations were associated with at least 2 fold increase in IC50
Major comments
1-Table 1: some of demographic data need to be added such as hepatocellular carcinoma, morbidity, viral load, presence of co-infection, etc. Also statistic analysis should be done in table 1
2-Figure 1: Did the authors submit the sequences in Genbank. If not please submit. Also please identify the amplicon size for the amplified HCV that used in phylogenetic tree.
3- Figure 2 and methodology are completely missing. First the methodology is not clear. I wonder why the authors did not construct all the previous 3 mutations in construct and test its replication efficiency.
4- According the study, did the authors identify the risk factor associated with emergence of these mutants.
Reviewer 2 Report
The work was carried out at the modern methodological level. Samples were identified using Next generation whole genome sequencing and bioinformatic analysis methods. A cluster of 132 cases infected with genotype 2a HCV was identified. As a result, the authors were able to show that lower response to treatment with glecaprevir/pibrentasvir was associated with the combination of L31M and R353K mutations within NS5A.
The authors used full-length genotype 2 HCV sequences published in ncbi for comparison, but did not themselves deposit the sequences described in this work. In order for the work not to be speculative, and its data to also become the property of the scientific community, it is necessary to place the sequences obtained during the work for cluster of 132 cases infected with genotype 2a HCV in the database or provide a link if these sequences have already been deposited. The current work only refers to one sequence isolated from a patient in 2013 (KY620322). In addition, a sequence KY320321 is erroneously indicated, the search for which in the NCBI leads to an unrelated sequence (Nitzschia dubiiformis strain SH366 ribulose-1,5-bisphosphate carboxylase/oxygenase large subunit (rbcL) gene, partial cds; chloroplast).
Comments
1) Design questions
- The authors have significantly exceeded the requirement for the design of the abstract, which says: “A single paragraph of about 200 words maximum”. Reframe the abstract to be around 200 instead of 290 words.
- The authors also incorrectly formatted references to literary sources, used round brackets, not square brackets (which is required by the terms of the journal).
- The bibliography is written using a font type other than the body text.
- The manuscript lacks page markings by line numbers, with the exception of the bibliography. This complicates the process of reviewing and feedback from the authors of the manuscript, since there is no way to attribute the comment to one or another markup line of text.
Abstract
2) “A cluster of 132 cases infected with genotype 2a HCV, were identified in the NHS Greater Glasgow & Clyde area.”
Correct to “was identified” (because you are talking about a cluster, in the singular).
3) “We identified the presence of L31M in”
You mention the L31M mutation for the first time in the abstract, but you do not specify in which protein it is. It might be better to write “mutation L31M in the viral protein NS5A” to make it clear to readers of the article what is being said.
Introduction
4) “As a result, HCV infection accounts for 350 to 500,000 deaths per year”.
The claimed spread of values, more than three orders of magnitude, looks too superficial for article 2022, even for a rough estimate. The fact is that the authors refer to a literary source from 2006, when there was no the introduction of direct-acting antivirals (DAAs). Please provide more recent estimates of the number of deaths from HCV infection per year, or do not provide these figures at all. They confuse with their degree of rough estimate.
Figure 1.
Figure caption
5) “this tree shows the subtype 2a cluster”
Start a new sentence with a capital letter
6) What is the difference between “Failed treatment” and “Not treated with G/P”? Also give a designation in the legend for cases that are not filled in the color boxes (white in Figure 1).
7) What is the scientific sense to compare on this phylogenetic tree sequences obtained, including from individuals that were not treated with glecaprevir and pibrentasvir (G/P) ? After all, Individuals that were not treated with glecaprevir and pibrentasvir (G/P) cannot be used for resistance statistics for such type of treatment.
Appendix A
8) Include in the legend not only the patients in the blue and green boxes (individuals from Glasgow and those outside of Glasgow), but also the non-boxed white sequences (for sequences downloaded from ncbi).
Round 2
Reviewer 1 Report
The authors addressed my questions